# Molecular Dynamic Simulations and Experiments Study on the Mechanical Properties of HTPE/PEG Interpenetrating Polymer Network (IPN) Binders

**DOI:** 10.3390/nano13020268

**Published:** 2023-01-08

**Authors:** La Shi, Xiaolong Fu, Yang Li, Shuxin Wu, Saiqin Meng, Jiangning Wang

**Affiliations:** Xi’an Modern Chemistry Research Institute, Xi’an 710065, China

**Keywords:** HTPE/PEG, IPN, mechanical properties, molecular dynamic simulation, crosslinking structures

## Abstract

The mechanical properties of HTPE/PEG interpenetrating polymer network (IPN) binders were systemically studied with molecular dynamics (MDs) simulations and experiments. In this study, an algorithm was used to construct the crosslinking interpenetrating polymer network models and then the mechanical behaviors of Hydroxyl-terminated polyethylene glycol-tetrahydrofuran co-polyether/poly ethylene glycol (HTPE/PEG) IPN models were analyzed at a molecular scale. Firstly, glass transition temperatures (T_g_), mean square displacement (MSD) and mechanical properties of IPN crosslinked model simulations showed that better thermomechanical parameters appeared at low temperatures, which were in good agreement with the experimental methods, including dynamic mechanical analysis and uniaxial tensile. Then bond-length distribution was performed to verify the crosslinked structures between prepolymers and curing agents. FTIR-ATR spectra analysis of four IPN binder specimens also gave a convictive result to the special interpenetrating polymer network of polyether polyurethane binders. Cohesive energy density and friction-free volume explained how the micro-structures of IPN crosslinked models and the force of inter-molecule chains affected the mechanical behaviors of the HTPE/PEG polyurethane matrix. Lastly, the morphology of IPN binder specimen tensile fracture indicated the mechanism at different temperatures. These studies were helpful in understanding the mechanical properties of HTPE/PEG interpenetrating polymer network binders and provide molecular insight into mechanisms of mechanical behaviors, which would guide the property improvement of HTPE propellant.

## 1. Introduction

Hydroxyl-terminated polyether (HTPE) solid propellant was the only composite solid propellant that had passed the low-insensitivity munition test [1,2]. However, the ultimate stress and break elongation of HTPE propellant couldn’t meet the needs of applications, and how to improve the mechanical properties of HTPE propellant has attracted much research. Interpenetrating polymer network (IPN) was a hybrid combination of two or more independently cross-linked and/or polymerized in the immediate presence of others, mutually penetrating polymer networks without chemical bonding, which preserved the structural characteristics of each network and improved the overall properties of the polymer [3,4]. For the mechanism of IPN formations, one phase network was formed in the presence of the other phase network, or both networks were formed simultaneously, which were widely used in functional materials, biomedical materials and armaments [5]. The throughout and entanglement forces of IPN polyurethane were stronger than the packing enhancement and normal mechanical blending reaction, which helped the stress of the materials evenly pass to the interlocking of two networks by macromolecular chains. Even with the existence of silver lines, IPN materials could also suppress the silver lines by interlocking network development so as to improve the mechanical properties of the blends.

Polyether polyurethane showed good processing technology, low-temperature flexibility and water resistance [6]. Applying the IPN technology to solid rocket propellants [7], which could screen various base polymers and compatible blending methods [8], improved the mechanical properties, the specific impulse [9] and the process performance. Glycidyl azide polymer (GAP) -hydroxyl terminated polyether (HTPE) semi-interpenetrating network via synchronous dual curing systems presented good tensile strength, breaking elongation and thermal stability [10]. Abbas et al. improved the mechanical properties by developing an energetic IPN, which combined the cross-linking a cyl-terminated glycidyl azide polymer (a cyl-GAP) and hydroxyl-terminated polybutadiene (HTPB) [11]. Poly(ethylene glycol) (PEG)and polymethyl methacrylate (PMMA) were synthesized by the sequential technique as semi-interpenetrating polymer networks (semi-IPN), which observed that mechanical properties of composite materials were improved with the increase of PMMA [12]. Poly(ethylene oxide-co-tetrahydrofuran) (PET) was embedded into the tridimensional network of triazole by the sequential-IPN process and these special materials showed good thermal stability and positive heat of formation [13]. The functionalized carbon nanotubes were introduced into HTPB and GAP, forming the IPNs, which could improve mechanical and thermal properties as binders for solid propellants [14]. Jian et al. [15] investigated that PEG could enhance the compatibility of PMMA/PU semi-IPNs and the that tensile stress of the composite materials could get to 9.6 MPa. Hydroxyl-terminated polyethylene glycol-tetrahydrofuran co-polyether (HTPE) was co-polymerized with HTPB and formed crosslinked polyurethane subsequently under the zinc chloride surrounding, presenting a low glass transition temperature and excellent mechanical properties including an elongation at break of 1501% and tensile strength of 8.02 MPa at 40 °C [16]. The blends of GAP and PET were crosslinked with toluene diisocyanate (TDI), which were introduced into the binders of crosslinked modified double base propellant appearing tensile strength of 1.20 MPa, elongation at break of 273.7% and a low glass transition temperature [17]. The nitrate ester plasticized polyether propellant (NEPE) composed of PEG binders could maintain well-viscoelastic behavior with the stretch and the dewetting phenomenon of the filler was not obvious while showing the drawing of the matrix and the growth of the holes [18]. GAP-based solid propellants showed enhanced mechanical and thermal properties with the incorporation of PEG [19], especially since the GAP/PEG propellant was much softer. PEG improved the compatibility of plasticizers with HTPB/PEG polyurethane matrices and increased the modulus and tensile strength of these IPN binders [20]. Additionally, graft semi-IPNs that were interpenetrated with PEG8000 and acrylic copolymers showed high thermal and cycling stability with excellent shape solidity and no change in crystallization properties compared to the pristine PEG8000 [21]. A tightly crosslinked PEG network was interpenetrated with a second network that subsequently swelled with an adjustable pH and then hydrogel materials that performed more perfect modulus and tensile stress properties would be obtained [22]. When the crosslink density of IPNs was small, the free energy would contribute a limited role to the mutual entanglement of the works and could be neglected [23]. The IPN matrix got high mechanical strength, which was composed of a chemically crosslinked poly(N,N-dimethylacrylamide-co-N-succinimidyl acrylate) network and a physically crosslinked poly(vinylidene fluoride-co-hexafluoropropylene) network [24]. In terms of these views, it’s meaningful that small molecular chains were crosslinked [25,26] in the immediate presence of HTPE by synthetic techniques, combined with molecular dynamic simulations and experimental methods [27].

In this study, HTPE and PEG prepolymers were used to crosslink with the four different curing agents, and two kinds of polyurethane structures would form interpenetrating polymer structures. Molecular dynamic simulations were carried out to explore the mechanical behaviors of four IPN models, including glass transition temperatures, bond-length distributions, mean square distributions, cohesive energy density and free friction volume. Furthermore, some essential experimental practices were used to identify the simulations. For example, ATR/FTIR would explain the chemical structures of IPN binders. Dynamic mechanical analysis not only implied the special structures of IPN binders but also gave a deep understanding of the thermomechanical properties of materials. Static tensile test and the morphology of IPN fracture could offer the mechanical properties and crack mechanism of different IPN binders.

## 2. Simulations

### 2.1. Model Constructions

In this paper, all molecular models and dynamic simulations were carried out by Materials Studio software, which could refer to the previous work [28]. The molecular structures of hydroxy-terminated polyether (HTPE), polyethylene glycol (PEG), polyfunctional isocyanate (N-100), hexamethylene diisocyanate (HDI), toluene diisocyanate (TDI), isophorone diisocyanate (IPDI), dibutyl phthalate (DBP) and four kinds of IPN models were displayed in Figure 1. Especially, the HTPE molecular chains were composed of polyethylene glycol and polytetrahydrofuran with a molar weight of 3974 g/mol. PEG400 molecules were the average of the PEG414 and PEG367 molecules. These polymers were built in the Builder Polymers module of Materials Studio.

The blended models were built in the Amorphous Cell module with the a 90.39 Å × 89.46 Å × 90.48 Å and the initial density was 0.6 g/cm^3^. The final density was determined by a geometry optimization. Then 10 frames of different curing systems were output, and the one with the lowest energy was chosen as the basic structure. The van der Waals selected the atom-based method, and electrostatic interactions chose the Ewald method. For these bulk volume systems, the simulation quality was set to 15.5 Å.

When all blended models were established, the geometry structures would relax with 50,000 steps of energy minimization. Then all models would be annealed from 600 K to 300 K. The temperature ramp was 20 K and consisted of 10 annealing recycles, of which a 200 ps dynamics simulation was performed in the NPT [29] ensemble, NHL thermostat and Berendsen barostat [30]. Furthermore, the first output frame (600 K) of the last recycle was selected as the primary model. All models were optimized and analyzed under the compass Ⅱ force [31] in this paper. S-N, S-H, S-T and S-I were referred to as the crosslinking structures of the N-100, HDI, TDI and IPDI curing systems, respectively.

### 2.2. Computation Details

In recent years, constructing crosslinking structures of polymer molecules attracted many scientists [32,33,34,35]. For the IPN models, PEG would penetrate HTPE randomly to form stable prepolymers with the help of plasticizers. The prepolymers would react with the curing agents in terms of the principle of addition polymerizations. The primary blended models were exposed to a crosslinking procedure to realize the crosslinked IPN models.

The flow chart of the crosslinking mechanism algorithm was shown in Figure 2a,b, which could be realized by MS through Perl scripts. The core idea was to increase the cutoff distances of efficient ranges between the special atoms of -OH groups and -NCO groups step by step to generate the crosslinking structures. Of course, there were some assumptions underlying these simulations. The equal chances of HTPE, PEG, DBP, N-100, HDI, TDI and IPDI molecular chains would be given to produce crosslinking structures. The -O-H and -N=C=O bonds broke and formed among reactive atoms, the of -NCO and -OH groups, when they met in the specified distance ranges. This algorithm would finish when it reached the maximum cutoff distance or the set conversation. Rather than putting the whole polyurethane into boxes directly, the random crosslinked structures based on atomic collision can really reflect the experimental samples. The reactive distance would increase from 3.5 Å to 14.5 Å with a ramp of 0.5 Å. The max conversion (α), system temperature (T), min reaction radius (R_min_), max reaction radius (R_min_) and other parameters could be reset flexibly.

When the crosslinking models were established, it was essential to create energy minimizations and dynamic optimizations for these models. The final densities of four crosslinked models were determined by geometry optimization. The densities of HTPE-PEG/N-100, HTPE-PEG/HDI, HTPE-PEG/TDI and HTPE-PEG/IPDI crosslinked interpenetrating polymer structure models were 1.077 g/cm^3^, 1.074 g/cm^3^, 1.077 g/cm^3^ and 1.068 g/cm^3^, respectively. In order to get crosslinking structures utterly relaxed, one more annealing procedure needs to be performed [36,37,38]. This operation’s progress could refer to the primary blended models’ building. Up until now, the final crosslinking model was built successfully, which would be used to compute the special properties of HTPE binders. The final curing conversations of HTPE-PEG/N-100, HTPE-PEG/HDI, HTPE-PEG/TDI and HTPE-PEG/IPDI were 95%, 90%, 90% and 92.5%, whose crosslinking models could be seen in Figure 2c.

## 3. Experimental

### 3.1. Materials

Herein, the main prepolymer, curing agents and some essential additives were listed. Hydroxy-terminated polyether (HTPE, #61) was bought from Liming Research & Design Institute of Chemistry Co., Ltd., whose molecular weights were 3974 g/mol, the polydispersity was 1.58 and the hydroxyl value was 4.729 mol/g × 10^4^. Polyethylene glycol (PEG; CP) was provided by Sinopharm Chemical Reagent Co., Ltd., and its average formula weight was 400 dalton. Polyfunctional isocyanate (N-100) was provided by the Xi’an modern chemistry research institute with 5.32 mmol/g of isocyanate group. Toluene diisocyanate (TDI; AR) was provided by Sinopharm Chemical Reagent Co., Ltd. Hexamethylene diisocyanate (HDI; 99% purity) and dibutyl phthalate (DBP; 99% purity) were bought from Aladdin Co., Ltd., Shanghai, China. Isophorone diisocyanate (IPDI; 99% purity) was provided by Shanghai Macklin Biochemical Co., Ltd., Shanghai, China. The ALT-402, as a water remover, was provided by Changde Ailite New Material Technology Co., Ltd., Longgang City, China. Triphenyl bismuth (TPB), as a catalyst, was bought from Tanyun Chemical Research Co., Ltd., Yingkou, China. After the four IPN binder systems had been cured and shaped in the constant temperature oven, the binders were cut into the standard size for chemical structure analysis, thermomechanical property tests and fracture morphology characteristics.

### 3.2. Preparation of IPN binders

Firstly, the prepolymer HTPE melted from a white lax state to a uniform, clear and transparent liquid at 85 °C for at least 5 h. The PEG was added to produce the interpenetrating polymer network in a ratio of 1:1 with HTPE molecule chains. Secondly, 2~3wt% of ALT-402 were added to the compounds of two prepolymers. Then the blends were mixed uniformly, which would be dried with the extra water in a vacuum oven at 60 °C for 72 h. Thirdly, the curing agents N-100, HDI, TDI and IPDI were instilled with a NCO/OH group with a molar ratio of 1.2, respectively. The triphenyl bismuth as a catalyst was added at 0.3wt%. Forth, all raw materials were mixed to disperse uniformly and the mixtures would be degassed for 30 min in the vacuum oven. Finally, the blends need to be slowly poured into two different molds, and then they would be put into a water bath incubator for more than 5 days at 70 °C until the curing reaction was completely finished between the prepolymers and curing agents. A simplified experimental route to manufacture HTPE/PEG binders is shown in Figure 3a. The final experimental samples of HTPE/PEG interpenetrating polymer structure binders can be seen in Figure 3b.

### 3.3. Apparatus and Methods

Dynamic mechanical analysis (DMA850, single cantilever) could test the dynamic mechanical properties of the S-N, S-H, S-T and S-I interpenetrating polymer network binders with the tensile clamp. The HTPE binder samples size were 35 mm × 13 mm × 3 mm. The scanning temperature range was −130 °C to 20 °C and the low temperature was controlled with liquid nitrogen. The system heating rate was 3 °C/min, and the loading frequency was 1 Hz. The final result was analyzed from three samples to reduce the accidental error.

When the binders had been cut into JANNAF dog bones (length 75 mm × narrow parallel width 4 mm × thickness 2 mm), the mechanical properties such as the ultimate stress and the strain at break could be estimated at −40 °C, −20 °C, 0 °C, 20 °C and 50 °C and the loading rate was 500 mm/min. Five samples were needed for each binder system, and the average values of the mechanical properties were gained from the measured results, which could minimize accidental error. The experimental details could be referred to the standard GB/T528-2009, and it was performed by a universal testing machine (AG-X plus, 5 kN, 200 N, Shimadzu, Japan).

An attenuated total reflected Fourier transformed infrared spectrometer (ATR/FTIR, TENSOR #27, Germany, Bruker Corporation) was used to analyze the chemical structure information of IPN crosslinking binders. Its spectral resolution was 0.5 cm^−1^ and provided a wavenumber range of 4000~400 cm^−1^.

A scanning electron microscope (SEM; Quanta 600 F, America FEI Corporation) could present a magnification for the breakage surface morphological inspections of IPN crosslinking binders at different temperatures. The accelerating voltage was 20 kV, the working distance was 10.2 mm and all microcrack surfaces needed to be coated with conductive gold films.

## 4. Results and Discussion

### 4.1. The Simulation Section

#### 4.1.1. Bond-Length Distribution

The formations of penetrating polymer networks made the polyurethane different from the linear pre-polymers such as structures and properties. The bond-length distributions of the prepolymers, curing agents and primary crosslinking structure binders, which eliminated the solvent, were measured and analyzed. According to the algorithm, the crosslinking structures were successfully established between the -OH groups of prepolymers and the -NCO groups of curing agents. The final crosslinking structures presented significant differences from the originally blended prepolymers, which resulted from the breaking and deformation of bonds between molecules. The bond-length distributions of four crosslinked polyurethane, prepolymers and curing agents were investigated with Monte Carlo sampling, and the many-body interaction models enable one to take into account more realistic atomic architecture models [39,40,41]. The four systems showed little difference from each other in Figure 4 because all systems were composed of the same ratio of HTPE and PEG. Only a few different small molecular curing agents caused a little effect.

Through collections and comparisons of the bond length between the original polymers and the final crosslinking structures, it was obvious that the -OH bonds and -NCO bonds disappeared, subsequently, many -NH bonds and -COO- bonds were produced. It is approved that crosslinked models were established successfully by the Perl scripts and the compatibility of IPN models was perfect.

#### 4.1.2. Glass Transition Temperature

When the crosslinking models of IPN binders were established under the algorithm, the blended models needed to be sufficiently equilibrated and carried out molecular dynamic simulation under the NPT ensemble. The temperature decreased from 450 K to 150 K in 10-K intervals. One frame was output every 250 steps, and each temperature gradient contained 50 picoseconds for a total of 50,000 steps. The lowest energy frame was chosen to collect the physical parameters, such as density, volume and so on. All IPN binder models were well optimized; the glass transition temperature procedure was trained five times, and the average density of each model was taken to eliminate accidental errors. The density versus temperature relationships were fitted in Figure 5, and the discontinuity point was the glass transition temperature. For the thermosetting polymers, vitrification indicated a transformation from a liquid or rubbery state to a hard or glassy state [42], and the molecules or atoms were mainly subject to vibration without translational and rotational motion at the glass transition temperature (T_g_) [43]. T_g_ was a vital parameter to reveal the physical properties of polymer molecules and it was supposed to improve the mechanical properties of materials around T_g_. From Figure 5, T_g_ from the S-N, S-H, S-T and S-I binder models were 226.59 K (−46.56 °C), 218.28 K (−54.87 °C), 220.13 K (−53.02 °C) and 227.78 K (−45.37 °C), respectively. All the simulations of T_g_ on the penetrating polymer structure models were lower than those of the pure HTPE and the pure PEG curing agent crosslinked models in our previous work [28,44]. The T_g_ of the S-H binder model was the lowest, and S-I showed the highest, which was due to the difference in curing agents. However, the HTPE prepolymers and plasticizer determined the performance of the binder matrix, and a few curing agents had an overlooked effect on the mechanical properties of IPN models. As a result, all curing models of IPN binders were kept at a similar glass transition temperature. The polyether and methylene content of the polyurethane soft parts led to low glass transition temperatures, and these binders were soft and flexible at room temperature [45,46]. The cooling rate of the procedure was higher than the scanning rate in the dynamic mechanical analyzer so the fitted T_g_ was slightly higher than the real ones [47].

#### 4.1.3. Mean Square Displacement (MSD)

As aforementioned, the IPN binder models emerged with better mechanical properties under low temperatures and had terrible mechanics when the temperature rose. The diffusivity and motion behavior of binder systems were connected to the mechanical properties of the polyurethane matrix and could be calculated from the mean-square displacement (MSD) of molecules. Of course, crosslinked structures showed less mobility than free molecular chains. It was necessary to explore the mechanical properties of IPN binder models at various temperatures. MSD determined the movements of atoms over time and was defined as Equation (1).
(1)$$\mathrm{MSD}\left(\tau \right)=\frac{1}{N}\overset{N}{\underset{i=1}{\sum }}\langle {\left|r\left(\tau \right)-r\left(0\right)\right|}^{2}\rangle$$
where N is the number of atoms, $r\left(\tau \right)$ an $r\left(0\right)$d indicate the position of an atom at time τ and the beginning time. 〈·〉 is the ensemble average, and it was NVT in this paper.

MSDs of IPN binder model analyses were illustrated at different temperatures in Figure 6. With the temperature increasing, the MSDs of four IPN models became larger. It indicated that the motions of crosslinked IPN models were significantly influenced by the temperature. When it came to the same time scales between the position of moving atoms and their original position, the MSD curve of low temperature tended to a constant, especially the ones around glass transition temperature. However, an obvious gap occurred from 273.15 K to 293.15 K when the simulated time increased and it confirmed the supposed mechanical properties at low temperatures. Four IPN binder models showed similar trends, which arose from the same pre-polymers and plasticizers occupying much space in the crosslinked structures and the curing agents only contributing a little.

#### 4.1.4. Mechanical Properties

As we all know, the mechanical behaviors of composite solid propellants were mainly influenced by the physical and chemical properties of pre-polymers, curing agents and the final integrity of crosslinked polyurethane. The constant strain method was used to estimate the elastic constant matrix by a series of finite difference approximations. In detail, a small strain, which was set at 0.003 and a number of strains was 6, was applied to the periodic models, and arose the model deform in the xy, xz and yz planes when an initial energy equilibrium minimization was completed. Based on the assumption of linear elasticity, the stress-strain relation could be summarized by the generalized Hooke’s law as Equation (2).
(2)$$\sigma_{\mathrm{ij}}{=C}_{\mathrm{ij}}\varepsilon_{\mathrm{ij}}$$
where $\sigma_{\mathrm{ij}}$ was the stress and $\varepsilon_{\mathrm{ij}}$ (i, j = 1, 2, 3) was the strain tensor. $C_{\mathrm{ij}}$ was the elastic constant and could be calculated by the $\left[C_{\mathrm{ij}}\right]$ stiffness matrix.

In the molecular dynamic simulation, stress could be calculated by the following Equation (3):(3)$$\sigma_{\mathrm{ij}}^{\alpha }=-\frac{1}{V^{\alpha }}\underset{\alpha }{\sum }\left(m^{\alpha }u_{i}^{\alpha }u_{j}^{\alpha }+\underset{\beta }{\sum }r_{i}^{\alpha \beta }f_{j}^{\alpha \beta }\right)$$
where $m^{\alpha }$, $u_{i}^{\alpha }$ and $u_{j}^{\alpha }$ were the mass, ith and jth velocity of the α atom; $V^{\alpha }$ is the volume of the finite blended system; $r_{i}^{\alpha \beta }$ and $f_{j}^{\alpha \beta }$ were the ith and jth force between α atom and β atom.

The IPN binders of solid propellants were isotropic polymers, so the effective Lame’s constants λ and μ could be calculated by Equations (4) and (5) in the stiffness matrix.
(4)$$\lambda =\frac{1}{6}\left(C_{12}{+C}_{13}{+C}_{21}{+C}_{23}{+C}_{31}{+C}_{32}\right)$$
(5)$$\mu =\frac{1}{3}\left(C_{44}{+C}_{55}{+C}_{66}\right)$$

Then, Young’s modulus E, shear modulus G, bulk modulus K and Poisson’ ratio v would be inferred from Equations (6)–(9).
(6)$$E=\frac{\mu \left(3\lambda +2\mu \right)}{\lambda +\mu }$$
(7)G=μ
(8)$$K=\lambda +\frac{2}{3}$$
(9)$$\upsilon =\frac{\frac{1}{2}\lambda }{\lambda +\mu }$$

The mechanical properties of four IPN binder systems were estimated at 73.15 K, 173.15 K, 273.15 K, 373.15 K and 473.15 K, respectively. The final results are shown in Figure 7, which were concluded from Equations (5)–(8). For polyether polymers, the H-bonded interactions could form physical entanglements of molecular chains, which enhanced the modulus and strength of the IPN binders. As expected, all HTPE/PEG interpenetrating polymer structure binder models demonstrated the best mechanical properties around the glass transition temperature and close mechanical parameters as a result of the prepolymer skeleton’s ability to hold a high percentage of solid loading. The mechanical properties of S-N, S-H and S-T crosslinking binders showed a similar trend with the temperature change, which reached their highest elastic modulus near the T_g_ and then tended to decrease. For the S-I binder model, it was obvious that the excellent elastic modulus appeared at the low temperature, which resulted from no movements among molecule chains beyond the T_g_. While the S-H and S-T binder systems presented the largest values of mechanical properties, the S-I binder system showed the lowest.

#### 4.1.5. Cohesive Energy Density

Cohesive energy (E_coh_) related to inter-molecule forces could be used to estimate the mechanical properties of IPN binders and provide more evidence to predict T_g_, which was described in Equation (10). The cohesive energy density (CED) of these IPN binder systems was the mean energy per unit volume if all inter-molecular forces were needed to remove all molecules to an infinite distance from each other and could be defined as Equation (11). Usually, a lower CED indicates a smaller T_g_ and a more prominent elastic modulus of the polymers.
(10)$$E_{\mathrm{coh}}=\langle E_{\mathrm{inter}}\rangle =\langle E_{\mathrm{total}}\rangle -\langle E_{\mathrm{intra}}\rangle$$
(11)$$\mathrm{CED}=\frac{E_{\mathrm{coh}}}{V}$$
where E_total_ was the total energy of the IPN binder system, E_inter_ was the total energy among all molecules and E_intra_ was the intra-molecule energy; V was the volume of the explored binder system. The bracket <…> was an average under an ensemble of NPT in this molecular dynamic simulation.

The CED of IPN binder models was calculated at temperatures of 73.15 K, 173.15 K, 273.15 K, 373.15 K and 473.15 K, which is shown in Figure 8. As we can see, the CED of all IPN binder models decreased linearly with the increasing temperature and stayed at a closed value at the same temperature. It was said that the inter-molecular force of HTPE/PEG IPN binder models would decrease with the rising temperature. Furthermore, the big molecule pre-polymers, HTPE and PEG, played a major role, and the small curing agents had an insignificant effect. The S-N IPN binder models always kept their scores higher than others. However, S-T, S-H and S-I IPN binder models performed a close value. It might be due to the special chemical structures of the curing agents, especially the three functional groups of N-100 and the ring structures of TDI and IPDI.

#### 4.1.6. Fraction Free Volume

Fraction free volume (FFV) was an important microscopic performance parameter associated with the structures of different curing agents and the migration of plasticizers, which could influence glass transition temperatures and the mechanical properties of the IPN binders. Usually, a smaller free volume implies a weaker movement of molecule chains and a higher elastic modulus. Investigating the free volume of blends with the molecular dynamic simulation method could provide a deep understanding of the 3D networks of interpenetrating polymer networks. Through free volume theory, the volume (V_T_) of liquid or solid substances is composed of two parts, one is the van der Waals volume occupied by molecules (V_0_) and the other one is the free volume (V_f_), interspace among the molecules. The visual distributions of V_0_ and V_f_ of four IPN models were displayed in Figure 9. Meanwhile, the specific values for all were shown in Table 1. The FFV could be calculated as Equation (12) with the Connolly surface in the Atom Volume and Surface modulus by the MD methods.
(12)$$\mathrm{FFV}=\frac{V_{f}}{V_{0}{+V}_{f}}\;\times \;100\%$$

In Figure 9, a little dispersed blue-black space in IPN models depends on the packing ability of molecules and the geometric constraints imposed by the interpenetrating polymer structures. Only a few differences were displayed, which were due to the small curing agents. The FFV values of S-N, S-H, S-T and S-I interpenetrating polymer structure binders were 12.02%, 12.24%, 12.73% and 12.80%, respectively. It was obvious that the pre-polymers and plasticizers determined the main space of all crosslinking models. The S-N binder model was the most compact as a result of its N-100 chemical structures. Meanwhile, the S-I model was predicted with the highest T_g_. Of course, it was in good agreement with the simulation result of T_g_.

### 4.2. The Experiments Section

#### 4.2.1. ATR/FTIR Analysis

Attenuated total reflected Fourier transformed infrared spectroscopy (ATR/FTIR) was used to characterize the formation of new bonds resulting from crosslinking structures, and the chromatogram analysis is shown in Figure 10. Possible infrared spectroscopy absorption of the polyurethane structures, attained from the curing reaction between the pre-polymers and the curing agents, could identify whether the interpenetrating network polymers were or not. There were obvious variations from the precursors to the primary polyurethanes, when the different curing binder systems were processed for enough time. Most absorption peaks in the ATR/FTIR spectra of crosslinking IPN binders were similar. A wide absorption band of amide -N-H groups stretching vibrations at 3337 cm^−1^ could be detected in all IPN specimens. The -C-H groups of IPN binders, pre-polymers and curing agents presented the stretching vibration peaks at 2950 cm^−1^ and 2856 cm^−1^. The absorption peak at 2270 cm^−1^, which was attributed to the asymmetric stretching vibration peak of -NCO, performed a large absorption intensity. Therefore, it indicated the most effective characteristic peak for the identification of -NCO groups, which proved that the carbamate structures had been created between the pre-polymers and curing agents. The symbol peaks of polyurethane could be observed at 1676 cm^−1^ and 1735 cm^−1^. The stretching vibration peaks of C-O bonds in the ester group were at 1230–1195 cm^−1^ and the stretching vibration peaks of C-O bonds in the ether group were around 1100 cm^−1^. However, a few differences in the various HTPE binders were also exhibited, such as the intensity of the main peaks, the position of peaks and even the number of peaks. These behaviors mainly resulted by the various kinds of curing agents, the extra curing agents and the final polyurethane structures. As already demonstrated by the ATR/FTIR spectra intensity, the new polyurethane groups could be significantly noticed and the structures of HTPE binders were successfully crosslinked according to the reaction schemes shown in Section 3.2.

#### 4.2.2. Dynamic Mechanical Analysis

In the dynamic mechanical analyzer (DMA) experiments, E’ was the storage modulus and indicated the elastic properties of materials, which mainly spring back to their pristine form after releasing the stress. E’’ was the loss modulus and represented the viscous properties of the materials, which disappeared as heat. Tan δ referred to the ratio of damping of E’’ to E’, the peak of which could reflect the glass transition temperature (T_g_) of the materials. As a result, the peaks of the tan δ curves were associated with the chain releasing of molecules under the development of the temperature. The results of E’, E’’ and tan δ as the function of temperature are shown in Figure 11.

As we can see, E’’ decreased with the increasing temperature and rapid damp was nearly the T_g_ in all systems. However, E’ showed a slight descent similar to E’’ at the beginning, an inverse rise to the peak and slid down later. The IPN binders were kept in a perfect elastic state under −80 °C because of a few damps below this temperature [48]. The crosslinking structures of IPN binders were extremely stable at these temperatures, even though no movements occurred among the molecular chains. From the curve peaks of tan δ, T_g_ of S-N, S-H, S-T and S-I interpenetrating polymer networks were seen at −69.28 °C, −72.18 °C, −69.24 °C and −68.67 °C, respectively. The concentrating T_g_ indicated that the mechanical properties of final IPN binders depended on the large molecular pre-polymers, and the small curing agents played an overlooked role. The signal-damping peaks of the tan δ curve for four crosslinked IPN binder systems could be attributed to some releasing of molecule chains with increasing temperature. For temperatures under −100 °C, the damping of the tan δ curve was quite low and it implied that the IPN binders stayed in a full elastic state without movements of molecular chains. In other words, DMA results provided distinguished significance regarding how the mechanical properties of different IPN binders were affected by temperature changes. Meanwhile, it verified that the mechanical properties of crosslinked IPN binder models could certainly reflect the micro-mechanism of mechanics by the molecular dynamic simulation method.

#### 4.2.3. Static Tensile Properties of IPN Binders

Curves of tensile stress versus strain could explain the relationship between tensile stress and tensile strain in IPN binder specimens’ tensile. Static tensile mechanical properties were necessary and important when studying the mechanical properties of polymer composites, especially for solid composite propellants. The stress-strain curves of S-N, S-H, S-T and S-I interpenetrating polymer network binder specimens tested at −40 °C, −20 °C, 0 °C, +20 °C and +50 °C were shown in Figure 12a–d, respectively. It was obvious that temperatures had a vital influence on the mechanical properties of IPN binders; the tensile strain at break increased with the decreasing temperature. Especially, the excellent tensile ultra-stress and strain at break suggested that the IPN binders combined the soft features of PEG and the high strength of HTPE polymers at low temperatures. Nevertheless, terrible mechanical properties were investigated at high temperatures. The mechanical properties of HTPE/PEG interpenetrating polymer structure binders were improved at high temperatures, compared with HTPE binders [49] and PEG binders. Maybe interrupted the force among inter-molecular chains of IPN binders so that the ultra-stress and strain at break were weakened a lot.

What is more, different from the curing IPN binder systems and the mechanical properties of IPN binders presented various rules with the increasing temperatures. For example, ultra-stress and strain at the break of the S-N binder specimen determined a similar value at 0 °C, +20 °C and +50 °C. However, a dramatical change appeared among 0 °C, −20 °C and −40 °C, which indicated that for the N-100 curing agent, the IPN binder specimens performed both the stress and strain at the break at −40 °C. For the crosslinked IPN binder of the S-H system, the ultra-stress at 0 °C and −40 °C was higher than others, while the highest strain at break was at 0 °C. Meanwhile, the mechanical properties of S-H binder specimens obviously varied with the temperature change. The ultra-stress of the S-T binder specimens displayed little difference around the high temperature but rose exponentially as the temperature went down from 0 °C to −40 °C. Similar to the S-H binder system, the values of strain at break were higher when the temperature was lower, and the highest strain at break of the S-T binder system was not at the lowest temperature. The ultra-stress of S-I binder specimens could be divided into three main values, belonging to 0 °C and −20 °C, +20 °C and +50 °C, or −40 °C. Although the values of strain at break tended to be larger at the lower temperature, the biggest one of the S-I binder specimens was at 0 °C. In other words, it could be observed how the contribution of the temperature and curing agent changing to the mechanical properties of four IPN binder systems. Of course, the measured data of IPN binder specimens also verified the simulation results of IPN binder models.

#### 4.2.4. Morphology

The failure of meso-mechanics of solid propellants mainly included three parts, wire-drawing and cavitation of the binder matrix, the de-wetting behaviors between the solid particles and binders and crystallite crushing. It was evident to explore the compatibility between the two mixed polyurethane networks acquired by HTPE and PEG reacting with curing agents through scanning electron microscopy (SEM). In this study, the tensile fracture surfaces of S-N, S-H, S-T and S-I interpenetrating polymer network specimens were observed in Figure 13a–d, numbered from one to five, which symbolized −40 °C, −20 °C, 0 °C, +20 °C and +50 °C. As seen, the tensile failure of all specimens was in valid positions. Some micro-phase separation morphology appeared as a result of the clear interface between the two phases, and these behaviors may be due to the interpenetration occurring in two different polyurethane matrices. The polyurethane composed of PEG and curing agents showed more urethane hard domains, while the other one contained more polyether soft chains through HTPE and curing agents. At low temperatures, a few silky stripes were seen in the polyurethane matrix, and this phenomenon arose from the formation of the polyurethane crystallites. The macroscopic break of IPN binder specimens hardly affected the state of hydrogen bonding or the hard phase network [50]. For the S-N binders, the roughness of the fracture surface varied with temperatures, and the ductile fracture behavior was obvious on the surface. The S-H binder systems presented a smoother surface of fracture, especially at micro-phase separation, which was significant at −40 °C and +50 °C. There was some spherical morphology in the S-T binders, which may result from PEG moieties at −40 °C. However, the surface of the fracture failure became softer, and a homogeneous film was filled with IPN binders. For S-I binder specimens, the crack morphology was coarser with a few agglomerations at +50 °C and the more even one with better compatibility at low temperatures. As mentioned in the analysis, both the temperatures and the curing agents played important roles in the fracture failure mechanism of IPN binders [51], which was in good agreement with the simulations.

## 5. Conclusions

In this paper, molecular dynamic simulations combined with experimental investigations were applied to study the mechanical properties of the HTPE-PEG interpenetrating polymer structures. Some interesting conclusions were as follows.

For the interpenetrating polymer structures, plasticizers played an important role, which could enhance the compatibility of different pre-polymers. The crosslinked models were built and used to predict mechanical behaviors at molecular scales. The bond-length distributions of IPN models were consistent with the FTIR-ATR spectra analysis of IPN binders. The glass transition temperatures of S-N, S-H, S-T and S-I binder models were 226.589 K, 218.283 K, 220.133 K and 227.777 K, respectively. The mean square displacements of four IPN models implied a weaker diffusivity and motion behavior of IPN models at low temperatures, which was connected to better mechanical properties. The mechanical properties of four IPN models were analyzed from 73.15 K to 473.15 K. It was proved that the more perfect mechanical properties of IPN binders around the glass transition temperature and worse elastic modulus appeared at 73.15 K or 473.15 K. Cohesive energy density and fraction-free volume was helpful in understanding how the micro-structures of IPN models affected the mechanical behaviors of materials. Meanwhile, dynamic mechanical analysis not only uncovered the interpenetrating polymer networks of two pre-polymers and curing agents, but also indicated that the storage modulus, loss modulus and loss factor varied with increasing temperature. What is more, the stress-strain curves of four IPN binders were analyzed by uniaxial tensile methods at −40 °C, −20 °C, 0 °C, +20 °C, +50 °C, respectively. All IPN binder systems showed the best mechanical properties at low temperatures and worse stress and strain at high temperatures. The morphology of fracture illustrated the micro-fracture mechanism of four IPN binders.

Overall, the S-I IPN crosslinked models showed the most stable structures and desired mechanical properties, which were also confirmed by the S-I binder specimens in experiments. From the analysis of four curing crosslinking systems, molecular dynamic simulations predicted the mechanical behaviors and explained the thermomechanical mechanism of IPN binders, which was in good agreement with experimental specimens.

## Figures and Tables

**Figure 1 nanomaterials-13-00268-f001:**
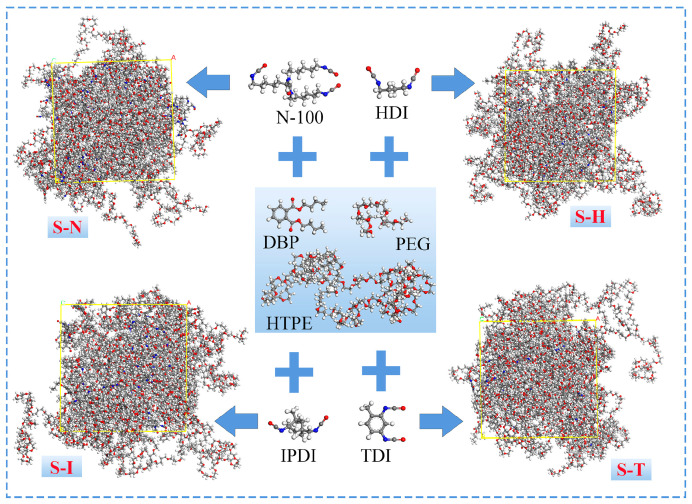
Molecules of prepolymers, curing agents, plasticizers and blended models of four systems.

**Figure 2 nanomaterials-13-00268-f002:**
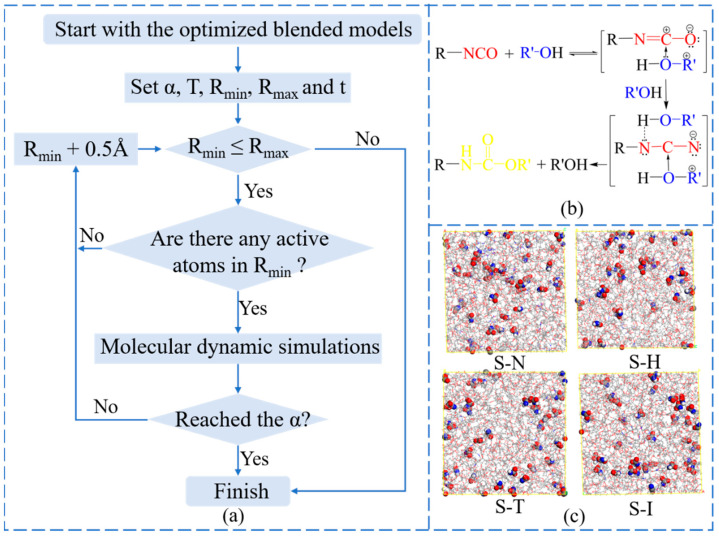
(**a**) Flow chart of the crosslinking reaction algorithm; (**b**) The curing reaction mechanism of IPN binders; (**c**) the crosslinking basic models of IPN binders.

**Figure 3 nanomaterials-13-00268-f003:**
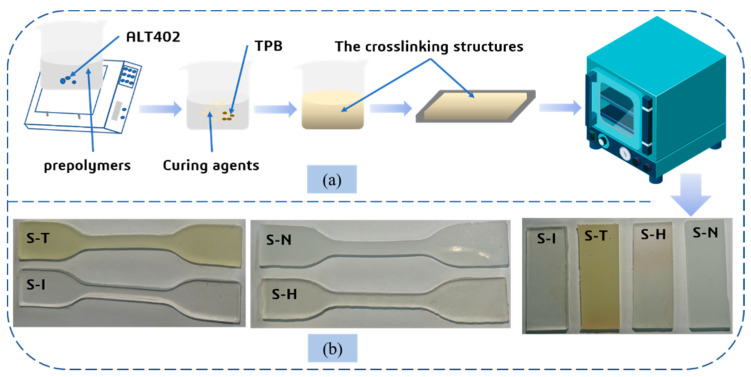
(**a**) the diagram illustration of the preparation process of INP binders; (**b**) the uniaxial tensile and DMA specimens of IPN binders.

**Figure 4 nanomaterials-13-00268-f004:**
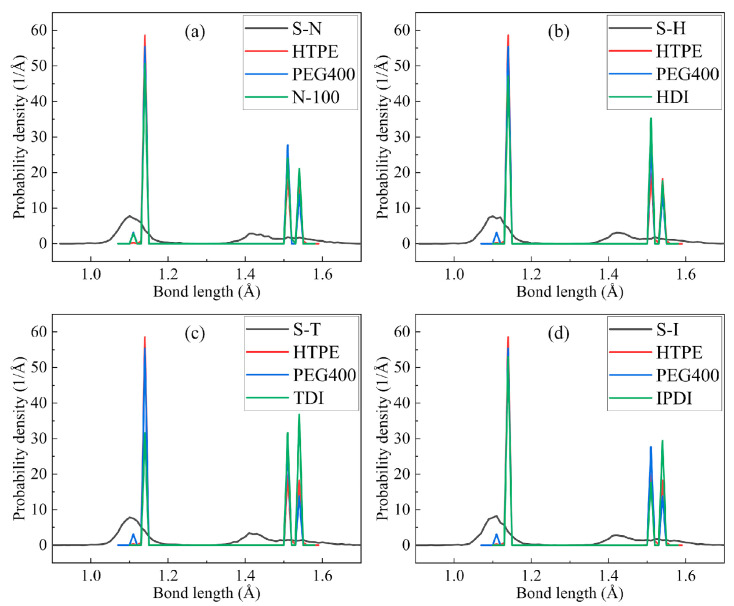
The bond-distributions of four interpenetrating polymer structure binder models. (**a**) S-N, (**b**) S-H, (**c**) S-T and (**d**) S-I.

**Figure 5 nanomaterials-13-00268-f005:**
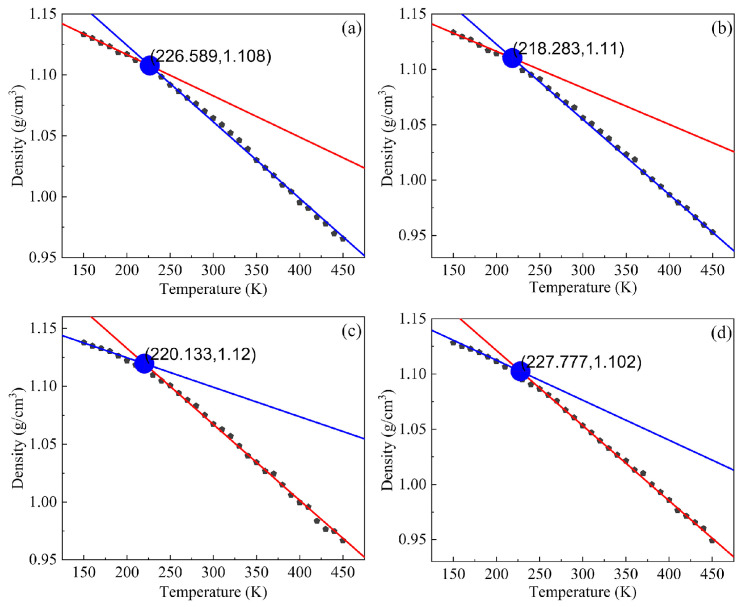
The glass transition temperatures of four interpenetrating polymer structure binder models. (**a**) S-N, (**b**) S-H, (**c**) S-T and (**d**) S-I.

**Figure 6 nanomaterials-13-00268-f006:**
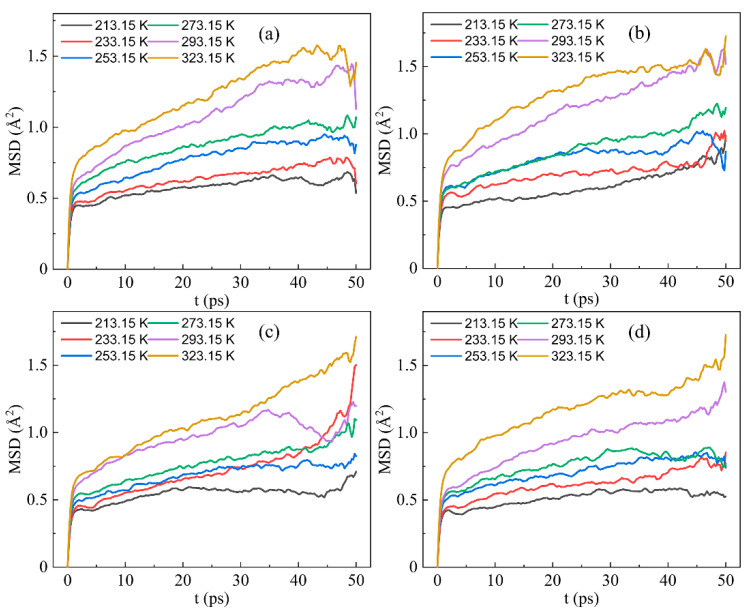
Mean squared displacement of four IPN binder models at 213.15 K, 233.15 K, 253.15 K, 273.15 K, 293.15 K and 323.15 K. (**a**) S-N, (**b**) S-H, (**c**) S-T and (**d**) S-I.

**Figure 7 nanomaterials-13-00268-f007:**
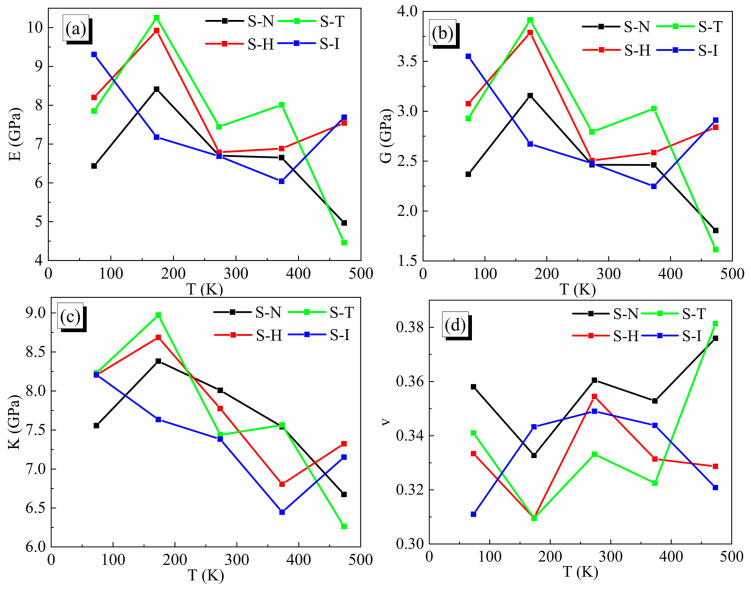
Mechanical properties of IPN binder models at 73.15 K, 173.15 K, 273.15 K, 373.15 K and 473.15 K. (**a**) Young’s modulus, (**b**) Shear modulus, (**c**) Bulk modulus and (**d**) Poisson’s ratio.

**Figure 8 nanomaterials-13-00268-f008:**
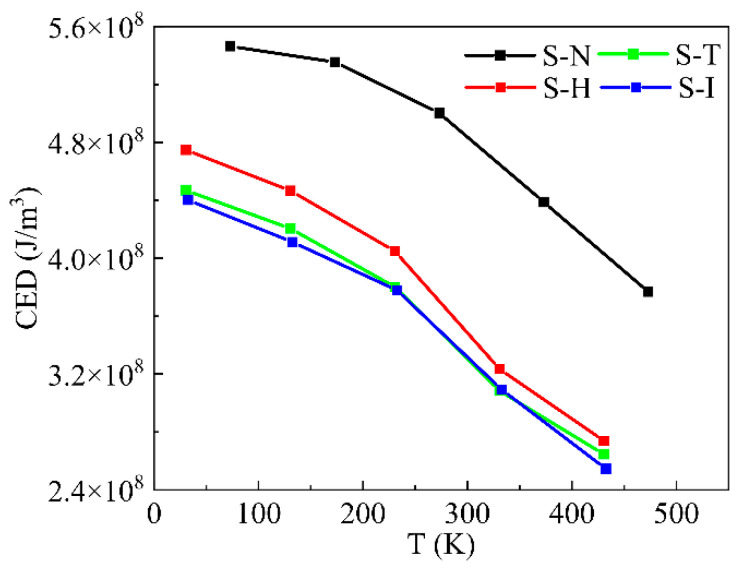
Cohesive energy density of IPN binder models at 73.15 K, 173.15 K, 273.15 K, 373.15 K and 473.15 K.

**Figure 9 nanomaterials-13-00268-f009:**
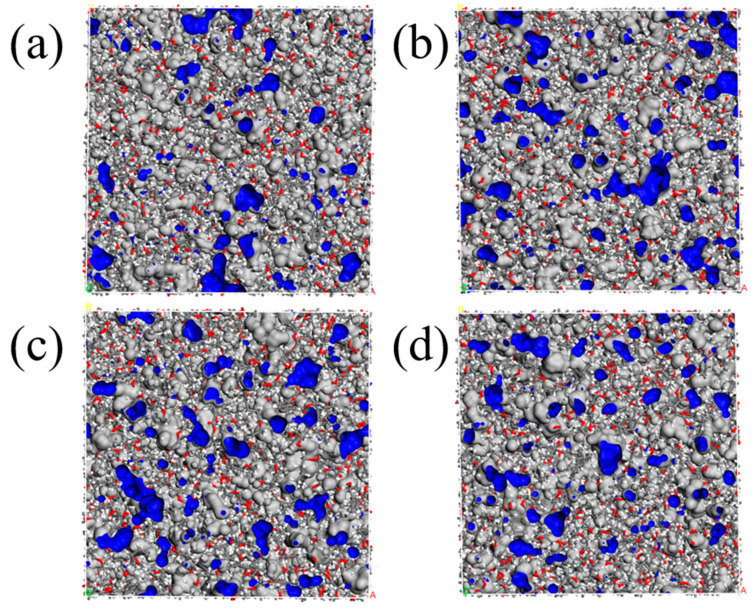
The free volumes of four crosslinking IPN binder models. (**a**) S-N, (**b**) S-H, (**c**) S-T and (**d**) S-I.

**Figure 10 nanomaterials-13-00268-f010:**
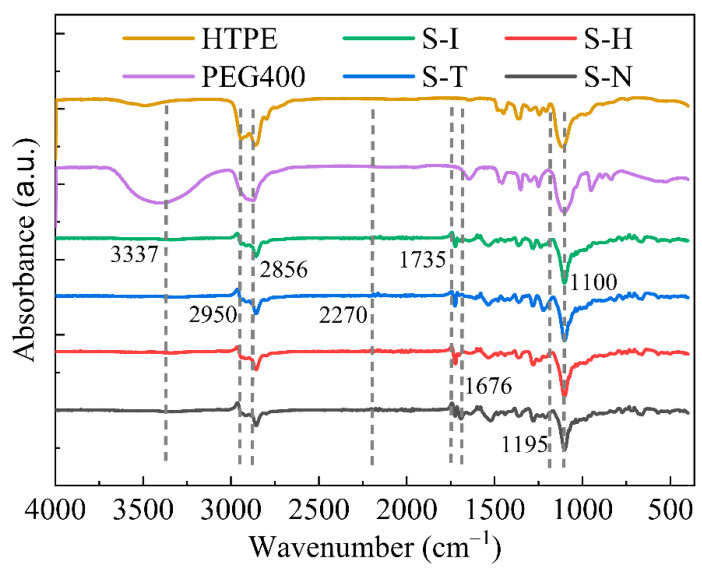
ATR/FTIR spectra of HTPE four HTPE/PEG interpenetrating polymer network binders.

**Figure 11 nanomaterials-13-00268-f011:**
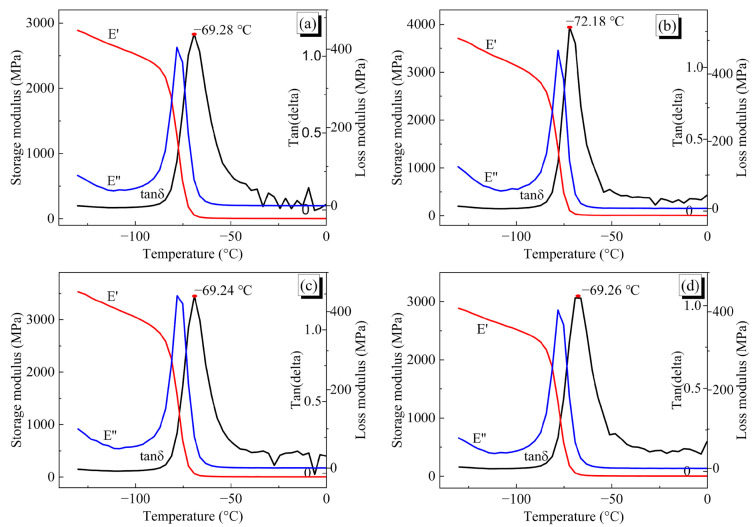
DMA curves of four HTPE/PEG interpenetrating polymer network binders. (**a**) S-N, (**b**) S-H, (**c**) S-T and (**d**) S-I.

**Figure 12 nanomaterials-13-00268-f012:**
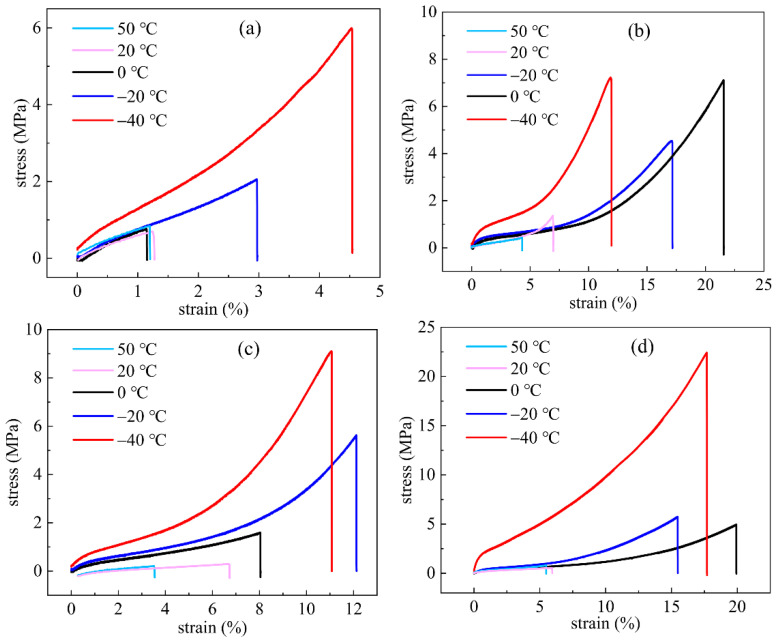
Curves of tensile stress versus strain of IPN binder specimens. (**a**) S-N, (**b**) S-H, (**c**) S-T and (**d**) S-I.

**Figure 13 nanomaterials-13-00268-f013:**
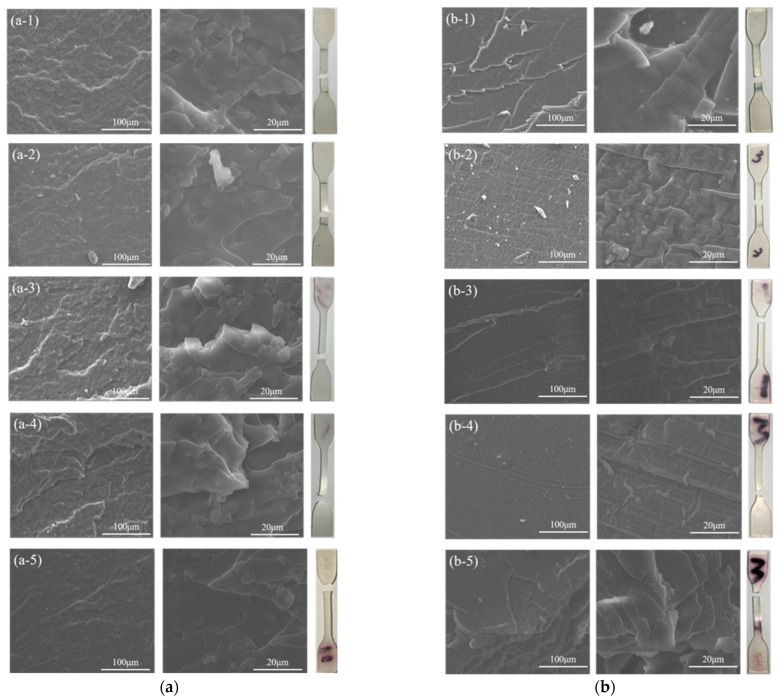
Scanning electron microphotographs of fracture of IPN matrix specimens. (**a**) S-N, (**b**) S-H, (**c**) S-T and (**d**) S-I. (**a**-**1**), (**a**-**2**), (**a**-**3**), (**a**-**4**) and (**a**-**5**) were the crack morphology of S-N binder specimen at −40 °C, −20 °C, 0 °C, +20 °C, +50 °C, respectively; (**b**-**1**), (**b**-**2**), (**b**-**3**), (**b**-**4**) and (**b**-**5**) were the crack morphology of S-H binder specimen at −40 °C, −20 °C, 0 °C, +20 °C, +50 °C, respectively; (**c**-**1**), (**c**-**2**), (**c**-**3**), (**c**-**4**) and (**c**-**5**) were the crack morphology of S-T binder specimen at −40 °C, −20 °C, 0 °C, +20 °C, +50 °C, respectively; (**d**-**1**), (**d**-**2**), (**d**-**3**), (**d**-**4**) and (**d**-**5**) were the crack morphology of S-I binder specimen at −40 °C, −20 °C, 0 °C, +20 °C, +50 °C, respectively.

**Table 1 nanomaterials-13-00268-t001:** The results of V_0_, V_f_ and FFV in HTPE/PEG interpenetrating polymer structure models.

System	V_0_ (Å^3^)	V_f_ (Å^3^)	FFV (%)
S-N	94,831.67	13,009.25	12.06
S-H	87,671.30	12,225.84	12.24
S-T	88,296.72	12,885.47	12.73
S-I	90,932.63	13,346.23	12.80

## Data Availability

Data available on request due to privacy.

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
