# Peer review of "Molecular Dynamic Simulations and Experiments Study on the Mechanical Properties of HTPE/PEG Interpenetrating Polymer Network (IPN) Binders"

_nanomaterials, 2023, doi:10.3390/nano13020268_

Round 1
Reviewer 1 Report
The manuscript "Molecular dynamic simulations and experiments Study on the mechanical properties of HTPE/PEG interpenetrating polymer network (IPN) binders" has an important and actual subject of the research field.
The work tresent molecular dynamic simulations combined with experimental investigations were applied to study the mechanical properties of HTPE-PEG interpenetrating polymer structures.
The general presentation of the work is good.
The methodologies, techniques and procedures were adequate.
The presented results are interesting and useful.
The design of the figures is good.
The conclusions were based on obtained results.
The references covering well the specific field of the research.
The manuscript is well prepared and only little polish of discussion is welcome.
The work can be considered for publish in NANOMATERIALS journal.
Author Response
Thank you for these suggestions and we have revised it carefully, which also could be seen in the revised manuscript.

Reviewer 2 Report
The article submitted by Fu. et al, entitled “Molecular dynamic simulations and experiments Study on the mechanical properties of HTPE/PEG interpenetrating polymer network (IPN) binders” (nanomaterials-2141395), investigated the mechanical properties of HTPE/PEG IPN binders using molecular dynamic simulations. The glass transition temperatures, bond-length distributions, mean square distributions, cohesive energy density and free friction volume were analyzed and some experiments were used to identify the simulations. The results suggest that the molecular dynamic simulation can help to understand the mechanisms of mechanical behaviors. Overall, the article is well-prepared, and it is recommended to be published. However, for the benefit of the reader, a revision is suggested before its publication.
In figure 13, the SEM pictures are better to be explained more clearly, and the details authors want to express can be marked in the figure.
Author Response

(The authors gave the same response as above.)

Reviewer 3 Report
The article "Molecular dynamic simulations and experiments Study on the mechanical properties of HTPE/PEG interpenetrating polymer network (IPN) binders"is interesting, but it needs improvement according to the recommendations below.
1. The introduction lacks emphasis on the main purpose of the research and references to works from recent years:
- Structure and Mechanism of Strength Enhancement in Interpenetrating Polymer Network Hydrogels, https://doi.org/10.1021/ma200693e
- Graft semi-interpenetrating polymer network phase change materials for thermal energy storage, https://doi.org/10.1021/acsapm.0c01363
- Novel Tough Ion-Gel-Based CO2 Separation Membrane with Interpenetrating Polymer Network Composed of Semicrystalline and Cross-Linkable Polymers, https://doi.org/10.1021/acs.iecr.1c04800
2.Please describe in detail how the final density was determined by a geometry optimization
3.In section:4.1.1. Bond-length distribution no literature reference.
4. . In section: Glass transition temperature, please provide the temperature in celsius and the relative value to pure polymer materials
5. Please summarize the mechanical properties in the table and discuss the trends in detail, because in their current form it is difficult to compare them.
6. The conclusions lack a summary and no selection of the most favorable polymer system network (IPN) binders
Author Response

(The authors gave the same response as above.)

Reviewer 4 Report
The authors have reported on the mechanical properties and cracking mechanisms of polymer structures with four types of crosslinkers. The agreement of experimental results from simulation predictions is interesting. However, there are some points that I would like to see revised. please refer.
major revision Unclear
1) Figure 3: Unclear
2) Figure 10: Unclear
3) Figure 11: Description of each line
4) Figure 13: Is it a magnified view of the same image? Is there a difference between a surface and a cross section? Please write a description of the range and temperature in the Figures session.
5) Line 144-151:Differences in curing occur. Are there many cases like this reported? I would like a reference.
6) Line 253-255 “the high prepolymer and plasticizer”: high・・・Unclear?:
minor revision
1) Figure 4. 5: Please add a description (a,b,c,d) to the figure, just like any other figure.
2) page 5:session number check
3) Line 160: 4.729 mol/g *104? "Is the exponent a number?" unit?
4) Line331: period check
Author Response

(The authors gave the same response as above.)

Round 2
Reviewer 4 Report
The authors have reported on the mechanical properties and cracking mechanisms of polymer structures with four types of crosslinkers. The agreement of experimental results from simulation predictions is interesting. I hope that this manuscript will be published in "Nanomaterials".